# The New Immuno-Oncology-Based Therapies and Their Perspectives in Hepatocellular Carcinoma

**DOI:** 10.3390/cancers13020238

**Published:** 2021-01-11

**Authors:** Philippe Merle

**Affiliations:** Centre de Recherche sur le Cancer de Lyon (CRCL), Hepatology and Gastroenterology Unit, Croix-Rousse Hospital, Hospices Civils de Lyon and INSERM U1052, Epigenetics and Epigenomics of Hepatocellular Carcinoma (EpiHep), 69004 Lyon, France; philippe.merle@inserm.fr; Tel.: +33-426109207

**Keywords:** hepatocellular carcinoma, systemic therapies, immuno-oncology

## Abstract

**Simple Summary:**

Hepatocellular carcinoma is a frequent and poor prognosis tumor, with most patients facing up, soon or later, to systemic therapies. So far, systemic therapies based on tyrosine kinase inhibitor monotherapies have been of modest benefit. The aim of this review article was to characterize the profile of efficacy and safety of immuno-oncology-based monotherapies that failed to demonstrate significant benefit, for comparison with the immuno-oncology-based combinational strategies. One of them has proven its drastic benefit in phase-3, whereas others have only shown promising data in phase-1/2, although the corresponding phase-3 results are pending. We showed that objective response rates and duration of response are important parameters for increased median overall survival and long survivals. We also pointed out that, being aware that there is an urgent unmet need for biomarkers, the pattern of safety and quality of life will guide the physician for the choice on the possible future combinations.

**Abstract:**

Hepatocellular carcinoma is a poor prognosis tumor. Systemic therapies are frequently used due to frequent recurrences after surgical or radiologic treatments. Anti-angiogenic tyrosine kinase inhibitors have shown efficacy in monotherapy, but with very low rates of long survival and exceptional recovery. Immuno-oncology based on immune checkpoint inhibitors has revolutionized the systemic therapies since showing long survival rates without any tumor progression or recurrence for some patients in partial or complete response, and possibly for some patients in stable disease. However, the rate of responders under immuno-oncology monotherapy is too low to increase significantly the median overall survival of the treated patients. The immuno-oncology-based combinations with different types of immune checkpoint inhibitors (PD-1/PD-L1 and CTLA-4 inhibitors such as nivolumab, pembrolizumab, atezolizumab, durvalumab, ipilimumab, tremelimumab), or the association of immune checkpoint inhibitors plus anti-angiogenic agents (bevacizumab, lenvatinib, cabozantinib), have led to a breakthrough in the treatment of hepatocellular carcinoma. Indeed, the first phase-3 trial, combining atezolizumab with bevacizumab, has dramatically changed the outcome of patients. Data from several other types of combinations assessed in phase-3 trials are pending, and if positive, will drastically arm the physicians to efficiently treat the patients, and disrupt the current algorithm of hepatocellular carcinoma treatment.

## 1. Introduction

Hepatocellular carcinoma (HCC) is one of the most prevalent malignant tumors with an incidence reaching more than 840,000 new cases per year and ranks fourth in terms of cancer-related death (around 740,000 deaths) worldwide [1]. The main risk factors are B (HBV) and C (HCV) chronic viral hepatitis, alcohol abuse, non-alcoholic fatty liver disease, and generally speaking any kind of chronic liver injury and more especially of cirrhosis [2]. Unfortunately, HCC diagnosis is commonly late while the tumor has spread outside the liver parenchyma as portal vein invasion or distant metastasis. In the history of HCC patients, a substantial proportion of them will have to face up, early or later, to systemic therapies due to no longer compatibility with radical or loco-regional therapies. Cytotoxic chemotherapies and hormone therapies have never shown any significant benefit on overall survival (OS) [3,4]. The first systemic therapy having demonstrated a significant beneficial impact on HCC outcome is sorafenib, a tyrosine kinase inhibitor (TKI) harboring anti-angiogenic and anti-proliferative properties on HCC [5]. During almost ten years, all the systemic therapies tested in randomized controlled trials in first-line systemic therapy head-to-head vs. sorafenib, or in second line after failure of sorafenib, have not shown any significant benefit—i.e., brivanib, everolimus, sorafenib plus erlotinib, tivantinib, sunitinib, anti-glypican-3 antibodies [6,7,8,9,10,11,12]. In 2018, Kudo et al. have shown that lenvatinib, another TKI with anti-angiogenic properties, was at least equivalent to sorafenib in first line in the REFLECT non-inferiority study [13]. In subgroup analysis for OS, lenvatinib tended to differ from sorafenib, with better efficacy (HR [95% CI]) in: (i) presence of macroscopic vein invasion and/or extrahepatic spread (HR 0.87 [0.73–1.04]) vs. absence (HR 1.05 [0.79–1.40]); (ii) high baseline level of alpha-fetoprotein (AFP) ≥ 200 ng/mL (HR 0.78 [0.63–0.98]) vs. low level < 200 ng/mL (HR 0.91 [0.74–1.12]); (iii) HBV-related HCC (HR 0.83 [0.68–1.02]) or HCV (HR 0.91 [0.66–1.26]) vs. non-viral (HR 1.03 [0.47–2.28]) [13].

Other TKIs with anti-angiogenic and anti-cancerous properties on HCC cells have demonstrated efficacy in second line: regorafenib after progression under sorafenib in 2017 [14], and cabozantinib after progression or intolerance under sorafenib strategy in 2018 [15]. In subgroup analysis for OS, regorafenib tended to differ from placebo, with better efficacy (HR [95% CI]) in: (i) presence of macrovascular invasion and/or extrahepatic disease (HR 0.63 [0.50–0.79]) vs. absence (HR 0.98 [0.58–1.66]); (ii) in presence of HBV (HR 0.58 [0.41–0.82]) vs. HCV (HR 0.79 [0.49–1.26]) vs. alcohol abuse (HR 0.92 (0.61–1.38]). Interestingly, no difference appeared regarding the baseline level of AFP: HR 0.68 (0.50–0.92) for high level ≥ 400 ng/mL vs. 0.67 (0.50–0.90) for low level <400 ng/mL [14]. The same trends were observed in subgroup analysis for OS regarding cabozantinib compared to placebo: (i) faint difference between high ≥ 400 ng/mL (HR 0.71 [0.54–0.94]) vs. low (HR 0.81 [0.62–1.04]) baseline AFP level; (ii) better efficacy when presence of macrovascular invasion and/or extrahepatic disease (HR 0.73 [0.60–0.90]) vs. absence (HR 0.99 [0.59–1.65]); and (iii) better efficacy in HBV-related HCC (HR 0.69 [0.51–0.94]), and by contrast to regorafenib its efficacy in alcohol (HR 0.72 [0.54–0.96]) and inefficacy in HCV (HR 1.11 [0.72–1.71]) [15].

Further, ramucirumab, a monoclonal antibody targeting vascular endothelial growth factor-2 (VEGFR-2), has shown significant benefit on a specific subpopulation of HCC patients, with high baseline AFP level ≥ 400 ng/mL [16]. In subgroup analysis for OS, ramucirumab tended to differ from placebo, with better efficacy (HR [95% CI]) in non-virus-related HCC (HR 0.63 [0.38–1.06]) vs. HCV (HR 0.76 [0.44–1.33]) vs. HBV (HR 0.84 [0.52–1.35]) [16].

However, all these options with TKI or ramucirumab monotherapies were strictly palliative with absence of long-term survivors and lack of potential recovery. Immuno-oncology approaches have completely revolutionized the paradigm of systemic therapies of HCC with nonetheless significant increase of median OS in combinational strategies, but also arising of possibility of long-term survivors and for some patients a hope of complete response and maybe definitive recovery. In this review article, we will focus on the advent of immuno-oncology currently assessed in phase-3 trials in the therapeutic strategies of HCC, the present standard of care of systemic therapy in first-line setting, and the potential strategies that will likely revolutionize the therapeutic algorithm of HCC therapy in a near future.

## 2. Limits of Immuno-Oncology as Monotherapy in Hepatocellular Carcinoma

### 2.1. Immune Checkpoint Inhibitors Targeting PD-1 Showed Promising Results in Phase-1/2 Trials

Among immune checkpoint inhibitors (ICI) of PD-1 that were further assessed in phase-3 trials for HCC, nivolumab was the first one to show encouraging data in the phase-1/2 CheckMate-040 trial [17]. The two main particularities by comparison with TKIs were the higher objective response rate (ORR) of 20% per Recist 1.1, and the long median duration of response (DOR) of 9.9 months (mo) (95% CI, 8.3-not evaluable [NE]). The ORR of TKIs per Recist 1.1 is rather low with sorafenib (2%) [5], regorafenib (7%) [14], and cabozantinib (4%) [15], and thus the median DOR remains hardly measurable with confidence due to the low number of events although a median value of 3.5 mo (95% CI, 1.9–4.5) was reported with regorafenib in the RESORCE trial [14], with these data being unavailable for sorafenib in SHARP [5] or cabozantinib in CELESTIAL [15]. However, this rational is different with the lenvatinib TKI, since giving a high level of ORR (18.8% per Recist 1.1 and 40.6% per mRecist), and DOR about 7.3 mo (95% CI, 5.6–7.7) [13].

The disease control rate (DCR) per Recist 1.1 of 64% with nivolumab tended to be equal or lower than with TKIs: sorafenib 71% [5], lenvatinib 72.8% [13], regorafenib 66% [14], and cabozantinib 64% [15]. Although OS is hardly assessable in phase-1/2 trials, nivolumab seemed to show interesting data since the median OS was not reached, being aware that the follow-up was short. On the other hand, progression-free survival (PFS) per Recist 1.1 did not seem to strikingly differ from TKIs (median, 95% CI): nivolumab 4.0 mo (2.9–5.4) [17], sorafenib 3.7 mo (3.6–4.6) in the control arm of the REFLECT trial [13] since unavailable in SHARP [5], regorafenib 3.4 mo (2.9–4.2) [14], cabozantinib 5.2 mo (4.0–5.5) [15], and lenvatinib 7.4 mo (6.9–8.8) [13].

Quite similar data were obtained with pembrolizumab in the phase-1/2 Keynote-224 trial [18] with ORR of 17%, median DOR not reached (95% CI, 3.1–14.6+), DCR of 62%, median OS of 12.9 mo (95% CI, 9.7–15.5), and median PFS of 4.9 mo (95% CI, 3.4–7.2). However, very interestingly with nivolumab but not assessed so far with pembrolizumab, ORR seemed to be a reliable surrogate marker of outcome since median OS (95% CI) was: (i) strikingly high for complete and partial responders (CR/PR) since not reached; (ii) low and similar to placebo groups in clinical trials for progressive diseases (PD) at 8.9 mo (7.3–13.4); and (iii) intermediate at 16.7 mo (13.8–20.2) for stable diseases (SD) [19]. Of notice is the appearance of a late flat tail in the Kaplan–Meier OS curves with immuno-oncology, which represents the long-term survivors. This late flat tail will be further discussed later on in data from phase-3 trials.

### 2.2. The Same Immune Checkpoint Inhibitors Targeting PD-1 Were Disappointing in Phase-3 Trials

Nivolumab and pembrolizumab monotherapies have been assessed in phase-3 trials, either in first line for nivolumab or in second line for pembrolizumab. CheckMate-459 is a phase-3 prospective open trial that randomized in first-line nivolumab to sorafenib in the control arm [20]. The nivolumab arm showed ORR of 15% per Recist 1.1, median time to response (TTR) of 3.3 mo (range, 1.6–19.4), median DOR of 23.3 mo (95% CI, 3.1–34.5+), and DCR of 55%. However, outcomes were disappointing since they did not statistically differ from the sorafenib arm, regarding: (i) OS of 16.4 mo (95% CI, 13.9–18.4) vs. 14.7 mo (95% CI, 11.9–17.2), HR 0.85 (95% CI: 0.72–1.02), *P* = 0.0752; and (ii) PFS of 3.7 mo (95% CI, 3.1–3.9) vs. 3.8 mo (95% CI, 3.7–4.5), HR 0.93 (95% CI, 0.79–1.10). However, the late flat tail of the OS Kaplan–Meier curves tended to be better in the nivolumab than in the sorafenib arm, with median follow-up of 15.2 mo (range, 0.0–38.8) and 13.4 mo (range, 0.1–38.4), respectively, between arms. Illogically, a late flat tail was present in the sorafenib arm. One of the possible explanations could be that almost half of the patients who received a subsequent therapy after sorafenib withdrawal were treated by immuno-oncology. Treatment effect on OS in predefined subsets showed a trend for better benefit in: (i) viral vs. non-viral-induced HCC with an unstratified HR (95% CI) of 0.77 (0.56–1.05) for HBV, 0.71 (0.49–1.01) for HCV, and 0.95 (0.74–1.22) for non-viral; (ii) presence (HR 0.74 [0.61–0.90]) vs. absence (HR 1.14 [0.81–1.62]) of macrovascular invasion and/or extrahepatic spread; and (iii) AFP with high baseline level ≥ 200 ng/mL (HR 0.69 [0.53–0.89]) vs. low level < 200 ng/mL (HR 0.99 [0.78–1.26]). Another remarkable data are the safety of nivolumab in comparison with sorafenib: fewer treatment-related adverse events (TRAE) of grade-3/4 (22% vs. 49% respectively), reason for discontinuation due to TRAE of 9% vs. 11%, and the clinically meaningful differences between treatment arms were observed for FACT Hep total in favor of nivolumab in terms of quality of life.

The same pattern of data was observed with pembrolizumab. Keynote-240 is a phase-3 prospective double-blinded trial that randomized in second-line pembrolizumab to placebo [21]. The pembrolizumab arm showed ORR of 18.3% per Recist 1.1, median TTR of 2.7 mo (range, 1.2–16.9), median DOR of 13.8 mo (95% CI, 1.5+–23.6+), and DCR of 62.2%. For nivolumab in CheckMate-459 [20], outcomes were disappointing, although with a strong trend for efficacy of pembrolizumab vs. placebo. However, the co-primary end-points did not reach the pre-specified *P* value required for statistical significance: (i) OS 13.9 mo (95% CI, 11.6–16.0) vs. 10.6 mo (95% CI, 8.3–13.5), HR 0.78 (95% CI, 0.61–0.99), *P* = 0.0238; and (ii) PFS 3.0 mo (95% CI, 2.8–4.1) vs. 2.8 mo (95% CI, 1.6–3.0), HR 0.72 (95% CI, 0.57–0.90), *P* = 0.0022. With nivolumab, the late flat tails of the OS Kaplan–Meier curves tended to be better in the pembrolizumab than in the placebo arm, with median (range) follow-up of 13.8 mo (0.9–30.4) and 10.6 mo (0.9–29.5), respectively. Such as CheckMate-459, illogically, a late flat tail was present in the placebo arm, and one of the main hypotheses is the administration of immuno-oncology in almost one-quarter of patients treated by subsequent anti-cancer therapy after placebo withdrawal. Subgroup analysis of OS showed a trend for better outcome (HR [95% CI]) in: (i) low baseline level of AFP < 200 ng/mL (HR 0.68 [0.49–0.96]) vs. high level (HR 0.88 [0.62–1.26]) by contrast with nivolumab in CheckMate-459; and (ii) in HBV-related HCC (HR 0.57 [0.35–0.94]) vs. HCV (HR 0.96 [0.48–1.92]) vs. non-viral (HR 0.88 [0.64–1.20]) in accordance with nivolumab in CheckMate-459 for HBV and non-viral subsets. Identically to nivolumab, not much TRAE of grade-3/4 occurred with pembrolizumab (18.3%), and reason for discontinuation due to TRAE was 6.5%. Furthermore, pembrolizumab preserved quality of life during treatment following the European Organization for Research and Treatment of Cancer Core Quality of Life Questionnaire (EORTC QLQ-C30) and its HCC supplement (EORTC QLQ-HCC18) [22].

## 3. Immuno-Oncology Combinations Raise Huge Hopes in the Treatment of Hepatocellular Carcinoma

### 3.1. The Atezolizumab/Bevacizumab Combination Has Become the Gold-Standard in First-Line Systemic Therapy

Prior to IMbrave-150 [23], the GO30140 phase-1b study led to exciting data about the anti-cancer synergy of atezolizumab and bevacizumab combination [24]. Bevacizumab, in addition to its therapeutic anti-angiogenic properties by inhibiting VEGF, silences the immunosuppressive tumor microenvironment and thus enhances atezolizumab efficacy [25,26,27]. The GO30140 study included five cohorts, with groups A and F being dedicated to HCC. In group A, all patients received atezolizumab (1200 mg) and bevacizumab (15 mg/kg) every 3 weeks. In group F, patients were randomly assigned to receive atezolizumab (1200 mg) plus bevacizumab (15 mg/kg) every 3 weeks, or atezolizumab (1200 mg every 3 weeks) alone. In group A, atezolizumab/bevacizumab combination, in comparison to immuno-oncology monotherapies with nivolumab [20] or pembrolizumab [21], tended to show much higher levels of ORR (36% per Recist 1.1) with the same long DOR (median not reached (95% CI, 11.8 mo–NE), and higher DCR (71%) with a median follow-up of 12.4 mo (IQR, 8.0–16.2). Outcomes seemed better with atezolizumab/bevacizumab [24] than nivolumab [20] or pembrolizumab [21]: (i) median OS of 17.1 mo (95% CI, 13.8–NE), (ii) PFS of 7.3 mo (95% CI, 5.4–9.9), and (iii) late flat tail of Kaplan–Meier curves at 33 mo. In group F, when comparing atezolizumab/bevacizumab with atezolizumab monotherapy, ORR was not much different (20% vs. 17% per Recist 1.1) with long median DOR (not reached [95% CI, NE] vs. not reached (95% CI, 3.7–NE), but higher DCR (67% vs. 49%) with a median follow-up of 6.6 mo (IQR, 5.5–8.5) for the atezolizumab/bevacizumab group and 6.7 mo (IQR, 4.2–8.2) for the atezolizumab monotherapy arm. The outcomes tended to be better in the atezolizumab/bevacizumab group: (i) median OS not reached in either treatment group (atezolizumab plus bevacizumab: 95% CI, 8.3 mo–NE; atezolizumab monotherapy: 95% CI, 8.2 mo–NE); and (ii) PFS of 5.6 mo (95% CI, 3.6–7.4) vs. 3.4 mo (95% CI, 1.9–5.2).

IMbrave-150 has raised the atezolizumab/bevacizumab combination as the gold-standard of HCC in first line [23]. This is a phase-3 prospective open trial that randomized in first-line atezolizumab (1200 mg) + bevacizumab (15 mg/kg) every 3 weeks to sorafenib in the control arm. Such as in group A of the GO30140 phase-1b trial, anti-cancer properties of the atezolizumab/bevacizumab arm were exciting with ORR of 27% per Recist 1.1, median DOR not reached, and DCR of 73.6% with a median follow-up of 8.6 mo at the time of the primary analysis (Table 1). Outcomes were encouraging vs. sorafenib with remarkable: (i) median OS not reached vs. 13.2 mo (95% CI, 10.4–NE), HR 0.58 (95% CI, 0.42–0.79), *P* = 0.0006; (ii) median PFS 6.8 mo (95% CI, 5.7–8.3) vs. 4.3 mo (95% CI, 4.0–5.6), HR 0.59 (95% CI, 0.47–0.76), *P* < 0.0001. The late flat tails of the OS Kaplan–Meier curves tended to be better in the atezolizumab/bevacizumab than in the sorafenib arm, but results are immature with a median follow-up of 8.6 mo at the time of the primary analysis (Table 2). Illogically, high median OS and presence of a late flat tail in the sorafenib arm, as discussed above for CheckMate-459 and Keynote-240, were possibly due to almost half of the patients receiving treatment after sorafenib withdrawal in the trial were treated by immuno-oncology. Subgroup analysis of OS showed a trend for better outcome (HR [95% CI]) in: (i) low baseline AFP level < 400 ng/mL (HR 0.52 [0.34–0.81]) vs. high ≥ 400 ng/mL (HR 0.68 [0.43–1.08]) in accordance with pembrolizumab in Keynote-240 but by contrast with nivolumab in CheckMate-459; (ii) presence (HR 0.55 [0.39–0.77]) vs. absence (HR 0.69 [0.29–1.65]) of vascular invasion and/or extrahepatic spread as observed with nivolumab in CheckMate-459; and (iii) HBV-related HCC (HR 0.51 [0.32–0.81]) and HCV (HR 0.43 [0.22–0.87]) vs. non-viral (HR 0.91 [0.52–1.60]) in accordance with nivolumab in CheckMate-459. Another remarkable data is the safety of atezolizumab/bevacizumab in comparison with sorafenib: fewer TRAE of grade-3/4 in 36% vs. 46%, respectively, with different patterns of AE, thus explaining a substantial delay to deterioration of patient-reported quality of life, physical functioning, and role functioning using the EORTC QLQ-C30. However, reason for discontinuation due to AE of any cause was 15.5% for atezolizumab/bevacizumab vs. 10.3% for sorafenib, although the same data from TRAE are unavailable (Table 3).

### 3.2. Other Types of Immuno-Oncology-Based Combinations Will Likely Compete with Atezolizumab/Bevacizumab in a Near Future in First-Line Setting

Other types of immuno-oncology-based combinations are ongoing in phase-3 trials, and results are pending with either PD-1/PD-L1 plus CTLA-4 inhibitors (nivolumab + ipilimumab in the CheckMate-9DW [NCT04039607], and durvalumab + tremelimumab in the HIMALAYA [NCT03298451]) or PD-1/PD-L1 inhibitors plus TKI (pembrolizumab + lenvatinib in the LEAP-002 [NCT03713593], and atezolizumab + cabozantinib in the COSMIC-312 [NCT03755791]. At the moment, we have available only data from phase-1/2 trials. Table 4 summarizes advantages and disadvantages of the different therapeutic strategies.

In the phase-1/2 Checkmate-040, a cohort assessed the combination of nivolumab plus ipilimumab [28]. Three schedules were tested and the most promising, in terms of ORR and OS, was NIVO1/IPI3: a first step of induction with nivolumab (1 mg/kg) + ipilimumab (3 mg/kg) every 3 weeks for 4 cycles, and a second step of consolidation with nivolumab (240 mg) every 2 weeks. Efficacy parameters were encouraging with ORR of 32% per Recist 1.1, median DOR not reached, median TTR of 2 mo (IQR, 1.3–2.7), but low DCR of 54% such as in immuno-oncology monotherapies (Table 1). Regarding outcomes, OS was 22.8 mo (IC 95%, 9.4–NE) (Table 2). As discussed with nivolumab monotherapy in CheckMate-040 [19], the pattern of ORR with nivolumab/ipilimumab appeared to be as a surrogate marker of OS with a median follow-up of 30.7 mo (range, 28.2–36.9): median OS not reached (95% CI, 33 mo–NE) for CR/PR, 14.5 mo (95% CI, 8.4–29.6) for SD, and 8.3 mo (95% CI, 6.6–10.8) for PD. Of notice is the late flat tail in the Kaplan–Meier OS curve at 36 mo. However, safety issues have emerged since the rate of grade-3/4 TRAE was 53%, and the percentage of patients discontinuing the treatment regimen because of study drug toxic effects was rather high at 22% (Table 3).

Regarding the ICI/ICI combination of durvalumab plus tremelimumab in the phase-1/2 STUDY-22 trial, durvalumab (1500 mg every 4 weeks) + tremelimumab (single injection of 300 mg) showed interesting data as well [29]. The ORR per Recist 1.1 was 24%, median DOR not reached with an unknown follow-up period (not reported), median TTR of 1.86 mo, and still a low DCR (45.3%) (Table 1) such as in nivolumab and pembrolizumab monotherapies [20,21], or nivolumab/ipilimumab combination [28]. The outcome parameters showed median OS of 18.7 mo (95% CI, 10.8–27.3), median PFS of 2.2 mo (95% CI, 1.9–5.4), and a late flat tail in the Kaplan–Meier OS curves at 28 mo (Table 2). Safety concerns seemed to be lower than with nivolumab/ipilimumab combination since the rate of grade-3/4 TRAE was lower (35.1% vs. 53%), as well as the rate of TRAE leading to treatment discontinuation (10.8% vs. 22%) (Table 3) [28,29].

The phase-1b STUDY-116 assessing the combination of a PD-1 inhibitor, pembrolizumab (200 mg every 3 weeks), with a TKI with high ORR, lenvatinib (8–12 mg/day) [13], highlighted the best results in terms of tumor response parameters [30]. Indeed, the ORR per Recist 1.1 was 36%, the median DOR of 12.6 mo (range, 6.9–NE), median TTR of 2.8 mo (range, 1.2–7.7), and DCR of 88% (Table 1), superior to DCR obtained with ICI monotherapies or ICI + ICI combinations [20,21,28,29]. The OS value was the highest and quite equal to that obtained with nivolumab/ipilimumab with a median value of 22 mo (95% CI, 20.4–NE), median PFS of 8.6 mo (95% CI, 7.1–9.7), and the late flat tail in the Kaplan–Meier OS curves at 30 mo (Table 2). As discussed with nivolumab monotherapy [20] and nivolumab/ipilimumab combination [28], ORR under pembrolizumab/lenvatinib combination seemed to be a surrogate marker on outcome with a median follow-up of 10.6 mo: median OS not reached for CR/PR (95% CI, 21.7–NE), 22 mo for SD (95% CI, 9.9–NE), and 2.3 mo for PD (95% CI, 1.4–4.6). Safety concerns seemed to be superior to those encountered with nivolumab/ipilimumab combination since the rate of grade-3/4 TRAE was 67% vs. 53%, although the rate of TRAE leading to treatment discontinuation was lower 14% vs. 22% (Table 3) [28,30].

Regarding the atezolizumab plus cabozantinib, no data from phase-1/2 trials are available whereas we dispose of data from the nivolumab (240 mg every 2 weeks) plus cabozantinib (40 mg/day) combination in the CheckMate-040 [31]. The ORR per Recist 1.1 tended to be lower than with other combinations (19%), a quite equal median DOR of 8.3 mo (range, 0.0–NE) with a median follow-up of 19.3 mo (range, 16.4–23.5), the median TTR being 4.8 mo (range, 2.7–20.7), and DCR of 75% (Table 1), quite equal to atezolizumab/bevacizumab combination [23]. The OS value was encouraging since reaching 21.5 mo (95% CI, 13.1–NE), median PFS of 5.4 mo (95% CI, 3.2–10.9), and the late flat tail in the Kaplan–Meier OS curves at 24 mo (Table 2). Safety concerns seemed to be better compared to pembrolizumab/lenvatinib combination with grade-3/4 TRAE incidence of 47% vs. 67%, and the rate of TRAE leading to treatment discontinuation 6% vs. 18% (Table 3) [30,31].

## 4. Conclusions and Perspectives

The phase-1/2 of immuno-oncology-based combination strategies have shown exciting data in the systemic treatment of advanced HCC, letting arising hopes for very long survivals and maybe even recovery for some patients. So far, only the IMbrave-150 trial has proven the superiority of atezolizumab/bevacizumab compared to a TKI, sorafenib. If the ongoing phase-3 trials become positive (CheckMate-9DW, HIMALAYA, LEAP-002, and COSMIC-312), hepato-oncologists will dispose of a large diversity of strategies in first-line setting. It is likely that the choice of a specific combination will be driven by its pattern of efficacy and tolerance, fitting with the profile of each patient. Of evidence, reliable biomarkers (tumor genetics or epigenetics, genetic polymorphism, immunophenotyping of HCC, …) predictive of efficacy of a specific immuno-oncology-based combination would be of huge help to make such a decision for personalized medicine, but none has been validated so far. At the moment, patients progressing under atezolizumab/bevacicumab combination are devoted to TKI monotherapy. However, clinical trials with immuno-oncology-based combinations will be soon carried out in second-line setting. Furthermore, immuno-oncology-based combinations might also change the therapeutic strategies of early HCCs in a neo-adjuvant and/or adjuvant setting, or improve the management of intermediate HCCs in association with, or in place of transarterial chemoembolization. Thus, the next coming years will be of great interest and might revolutionize the treatment of HCC and transform the poor prognosis of this disease.

## Figures and Tables

**Table 1 cancers-13-00238-t001:** Summary of response, disease control, and durability in the immuno-oncology-based combination trials.

Parameters	Nivo1/Ipi3	Durva/Treme300	Atezo/Beva	Pembro/Lenva	Nivo/Cabo
Median follow-up	30.7 mo (range, 28.2–36.9)	Not available	8.6 mo (primary analysis)	10.6 mo	19.3 mo (range, 16.4–23.5)
ORR per Recist 1.1	32%	24%	27%	36%	19%
Median DOR per Recist 1.1	Not reached	Not reached	Not reached	12.6 mo (range, 6.9–NE)	8.3 mo (range, 0.0–NE)
Median TTR per Recist 1.1	2 mo (IQR, 1.3–2.7)	1.86 mo	Not available	2.8 mo (range, 1.2–7.7)	4.8 mo (range, 2.7–20.7)
DCR per Recist 1.1	54%	45.30%	73.60%	88%	75%
Trial	CheckMate-040	STUDY-22	IMbrave-150	STUDY-116	CheckMate-040
Phase	1/2	1/2	3	1b	1/2
[Ref]	[28]	[29]	[23]	[30]	[31]

mo, month; ORR, objective response rate; DOR, duration of response; NE, not estimable; TTR, time to response; IQR, interquartile range; DCR, disease control rate.

**Table 2 cancers-13-00238-t002:** Summary of survival parameters in the immuno-oncology-based combination trials.

Survivals	Nivo1/Ipi3	Durva/Treme300	Atezo/Beva	Pembro/Lenva	Nivo/Cabo
Median follow-up	30.7 mo (range, 28.2–36.9)	Not available	8.6 mo (primary analysis)	10.6 mo	19.3 mo (range, 16.4–23.5)
Median OS (95% CI)	22.8 mo (9.4–NE)	18.7 mo (10.8–27.3)	Not reached	22 mo (20.4-NE)	21.5 mo (13.1–NE)
Median PFS per Recist 1.1 (95% CI)	Not available	2.2 mo (1.9–5.4)	6.8 mo (5.7–8.3)	8.6 mo (7.1–9.7)	5.4 mo (3.2–10.9)
Trial	CheckMate-040	STUDY-22	Imbrave-150	STUDY-116	CheckMate-040
Phase	1/2	1/2	3	1b	1/2
[Ref]	[28]	[29]	[23]	[30]	[31]

mo, month; OS, overall survival; NE, not estimable; PFS, progression-free survival; K-M, Kaplan–Meier.

**Table 3 cancers-13-00238-t003:** Safety profiles in the immuno-oncology-based combination trials.

Parameters	Nivo1/Ipi3	Durva/Treme300	Atezo/Beva	Pembro/Lenva	Nivo/Cabo
TRAE of grade-3/4	53%	35.10%	36%	67%	47%
Treatment discontinuation due to AE	22% (due to TRAE)	10.8% (due to TRAE)	15.5% (due to AE from any cause) †	14% (due to TRAE)	6% (due to TRAE)
Trial	CheckMate-040	STUDY-22	Imbrave-150	STUDY-116	CheckMate-040
Phase	1/2	1/2	3	1b	1/2
[Ref]	[28]	[29]	[23]	[30]	[31]

TRAE, treatment-related adverse event; AE, adverse event. † in IMbrave-150 the rate of discontinuation due to TRAE in not available.

**Table 4 cancers-13-00238-t004:** Comparisons of tyrosine kinase inhibitor (TKI) or immune checkpoint inhibitors (ICI) monotherapies, and immuno-oncology-based combinations from phase-3 (SHARP [5], RESORCE [14], CELESTIAL [15], REFLECT [13], CheckMate-459 [20], Keynote-240 [21], IMbrave-150 [23]), or phase-1/2 with near coming phase-3 studies (STUDY-116 [30], CheckMate-040 [28], STUDY-22 [29]).

Parameters	Advantages	Disadvantages
Sorafenib, regorafenib, cabozantinib	Oral pills DCR = high (64% to 71%) PFS = moderate for cabozantinib (5.2 mo) Treatment withdrawal rate due to TRAE = moderate (10% to 16%)	ORR = low (2% to 7%) DOR unknown or short (3.5 mo) PFS = short for sorafenib (3.6 mo) and regorafenib (3.4 mo) Median OS = modest (10.2 to 10.7 mo) Rare long-term survivorsT RAE grade-3/4 rate = high (49% to 50%); only any cause AE grade-3/4 available for cabozantinib (68%)
Lenvatinib	Oral pills ORR = intermediate (18.8%) DCR = high (72.8%) PFS = long (7.3 mo) Treatment withdrawal rate due to TRAE = low (9%)	DOR = moderate (7.3 mo) Median OS = modest (13.6 mo) Rare long-term survivors TRAE grade-3/4 rate = high (57%)
Nivolumab, pembrolizumab	ORR = intermediate (15% to 18.3%) DOR = long (13.8 to 23.3 mo) Significant proportion of long-term survivors TRAE grade-3/4 rate = low (18.3% to 22%) Treatment withdrawal rate due to TRAE = low (6.5% to 9%) Better quality of life vs. sorafenib or placebo	Intra-venous infusion in hospitalization DCR = low (55% to 62.2%) PFS = short (3.0 to 3.7 mo) Median OS = moderate (13.9 to 16.4 mo)
Atezolizumab/bevacizumab	ORR = high (27%) DOR = long (median not reached) DCR = high (73.6%) PFS = long (6.8 mo) Median OS = long (not reached) Significant proportion of long-term survivors Treatment withdrawal rate due to any AE is moderate (15.5%) but missing data related to TRAE Better quality of life vs. sorafenib	Intra-venous infusion in hospitalization TRAE grade-3/4 rate = intermediate (36%)
Pembrolizumab/lenvatinib	ORR = high (36%) DOR = long (12.6 mo) DCR = very high (88%) PFS = long (8.6 mo) Median OS = long (22 mo) Significant proportion of long-term survivors Treatment withdrawal rate due to TRAE = moderate (14%)	Intra-venous infusion in hospitalization (in addition to oral pills) TRAE grade-3/4 = high (67%)
Nivolumab/ipilimumab	ORR = high (32%) DOR = long (median not reached) Median OS = long (22.8 mo) Significant proportion of long-term survivors	Intra-venous infusion in hospitalization DCR = low (54%) PFS = not available TRAE grade-3/4 = high (63%) Treatment withdrawal rate due to TRAE = high (22%)
Durvalumab/tremelimumab	ORR = high (24%) DOR = long (median not reached) Median OS = long (18.7 mo) Significant proportion of long-term survivors Treatment withdrawal rate due to TRAE = low (10.8%)	Intra-venous infusion in hospitalization DCR = low (45.3%) PFS = short (2.2 mo) TRAE grade-3/4 = intermediate (35.1%)

ORR, objective response rate (assessed per Recist 1.1) as well as PFS, progression-free survival and DRC, disease control rate; TRAE, treatment-related adverse events; AE, adverse events.

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
