# Peer review of "The New Immuno-Oncology-Based Therapies and Their Perspectives in Hepatocellular Carcinoma"

_cancers, 2021, doi:10.3390/cancers13020238_

Round 1
Reviewer 1 Report
Comments:
Title: The new immune-oncology-based therapies and their perspectives in hepatocellular carcinoma
Merle P. presents a well written review on Immuno-oncology perspectives on the cure of Hepatocellular carcinoma (HCC). This cancer represents one of the most challenge tumor to cure to date.
The author pointed out the disappointing, for example, for results in Phase 3 for immune checkpoint inhibitors targeting PD-1 compared to Phase 1 and 2. This aspects are not often described so it important summarize these results. This review present in a clear way the results of different clinical trials.
Strength of the review is the presentation of results in combinatorial therapy of atezolizumab/bevacizumab. Clear representation of combinatory therapy data and precise explanation of results is evidenced.
This review differs from other numerous reviews on IO approach on HCC. More experimental approaches in-vitro models are elucidated and still in phase of basic research studies. Gastroenterologists and liver cancer researchers will find in this well written review interesting aspects on combinatorial approaches for IO in HCC.
I support the publication of this review.
Minor Comments:
• Introduction – it should expanded the section on HCC diffusion, including the numbers worldwide and including if possible short introduction on the risk factors of this disease.
• Conclusions – It could be important to include author considerations on the emerging concept of “personalized” therapy (already applied in some type of cancer), based also in the genetic background of the patient, with the combinatorial approach. Can they co-exist? Are they compatible?
Author Response
Dear reviewer,
Thank you for your comments.
Please find answers to your criticisms:
1) In introduction, I added informations on incidence, death, and risk factors.
2) In conclusion, I had already noticed that some predictive markers are urgently needed but unfortunately not identified and validated so far. However, following your remarks, I have added a sentence specifying that genetics, epigenetics or genetic polymorphisms, immunophenotyping of HCC have not shown any usefulness for personalized medicine in HCC so far.
Reviewer 2 Report
This is a much technical report which is likely to raise interest among a quite limited audience. The impressive number of abbreviations makes reading quite difficult. I would suggest commenting on sex disparity in HCC. This is a male-predominant disease and, nevertheless, we do not seem to have learnt the lesson of personalized sex medicine. Maybe Professor Merle may be willing to comment on this.
Author Response
Dear reviewer,
Thank you for your comments.
Please find answers to yours criticisms:
1) Yes there are many abbreviations. I tried to delete some of them as shown in the revised manuscript : 1L, 2L, IO, NR.
2) Yes there is a male predominance in HCC (sex ratio around 4:1) and mechanisms are poorly understood either basically (adiponectine, serotonine, gut microbiota, hormonal regulation of key cellular processes, sex steroid receptor expression,...) or epidemiologically in populations of patients in terms of absence of difference for exposure to risk factors. The purpose of this paper is not to discuss about sex-related hepatocarcinogenesis, and it would be out off the subject.
We can only argue that men do not respond differently to anti-cancer therapy by comparison with women, even in the (negative) hormono-therapy trials, and thus, personalized sex-medicine has not got any rationale so far in HCC. I don't see any interest to address these comments in the paper since off subject, except if required by the editor.
Reviewer 3 Report
Hepatocellular carcinoma (HCC) is a poor prognosis tumor. Systemic therapies are frequently used due to frequent recurrences after surgical or radiologic treatments. In this manuscript, the author reviewed the efficacy and recent clinical trials of anti-angiogenic tyrosine kinase inhibitors (TKI) monotherapy, immuno-oncology (based on immune checkpoint inhibitors) monotherapy, and the immuno-oncology-based combinations with different types of immune checkpoint inhibitors (PD-1/L1 and CTLA-4 inhibitors such as nivolumab, pembrolizumab, atezolizumab, durvalumab, ipilimumab, tremelimumab), or the association of immune checkpoint inhibitors plus anti-angiogenic agents (bevacizumab, lenvatinib, cabozantinib). The immuno-oncology-based combinations with different types of immune checkpoint inhibitors or the association of immune checkpoint inhibitors plus anti-angiogenic agents have led to a breakthrough in the HCC treatment. Data from several other types of combinations assessed in phase-3 trials will drastically arm the physicians to efficiently treat the HCC patients, and disrupt the current algorithm of hepatocellular carcinoma treatment.
This is a comprehensive review on the new immuno-oncology-based therapies and their perspectives in hepatocellular carcinoma. The manuscript was well prepared. This review can provide useful information for the clinicians to manage the HCC patients.
Comment
The authors were suggested to add a Table to compare the advantages and disadvantages of TKI monotherapy, immuno-oncology monotherapy, and the immuno-oncology-based combination therapy.
Author Response
Dear Reviewer,
Thank you for your comments.
I have added a Table 4 showing advantages and disadvantages of TKIs, IO monotherapies and IO-based combinational strategies.
Best regards
Philippe Merle